

# Sub-milliSievert ultralow-dose CT colonography with iterative model reconstruction technique

Lukas Lambert[1], Petr Ourednicek[2], Jan Briza[3], Walter Giepmans[4], Jiri Jahoda[1], Lukas Hruska[2] and Jan Danes[1]

[1] Department of Radiology, First Faculty of Medicine, Charles University in Prague and General University Hospital in Prague, Prague, Czech Republic
[2] Department of Imaging Methods, St. Anne's University Hospital in Brno, Brno, Czech Republic
[3] First Department of Surgery, First Faculty of Medicine, Charles University in Prague and General University Hospital in Prague, Prague, Czech Republic
[4] Clinical Science & Application Computed Tomography, Philips Healthcare, Best, The Netherlands

Corresponding author
Lukas Lambert,
lambert.lukas@gmail.com

## ABSTRACT

**Purpose.** The purpose of this study was to evaluate the technical and diagnostic performance of sub-milliSievert ultralow-dose (ULD) CT colonograpy (CTC) in the detection of colonic and extracolonic lesions.

**Materials and Methods.** CTC with standard dose (SD) and ULD acquisitions of 64 matched patients, half of them with colonic findings, were reconstructed with filtered back projection (FBP), hybrid (HIR) and iterative model reconstruction techniques (IMR). Image noise in six colonic segments, in the left psoas muscle and aorta were measured. Image quality of the left adrenal gland and of the colon in the endoscopic and 2D view was rated on a five point Likert scale by two observers, who also completed the reading of CTC for colonic and extracolonic findings.

**Results.** The mean radiation dose estimate was $4.1 \pm 1.4$ mSv for SD and $0.86 \pm 0.17$ mSv for ULD for both positions ($p < 0.0001$). In ULD-IMR, SD-IMR and SD-HIR, the endoluminal noise was decreased in all colonic segments compared to SD-FBP ($p < 0.001$). There were 27 small (6–9 mm) and 17 large ($\geq 10$ mm) colonic lesions that were classified as sessile polyps ($n = 38$), flat lesions ($n = 3$), or as a mass ($n = 3$). Per patient sensitivity and specificity were 0.82 and 0.93 for ULD-FBP, 0.97 and 0.97 for ULD-HIR, 0.97 and 1.0 for ULD-IMR. Per polyp sensitivity was 0.84 for ULD-FBP, 0.98 for ULD-HIR, 0.98 for ULD-IMR. Significantly less extracolonic findings were detected in ULD-FBP and ULD-HIR, but in the E4 category by C-RADS (potentially important findings), the detection was similar.

**Conclusion.** Both HIR and IMR are suitable for sub-milliSievert ULD CTC without sacrificing diagnostic performance of the study.

## INTRODUCTION

Over the last decade, we have witnessed substantial improvements in the iterative reconstruction technique that ultimately resulted in introduction of iterative model reconstruction (IMR) technique into practice by major CT vendors (*Löve et al., 2013*). Compared

to filtered back projection (FBP), which is a standard single-pass analytical method for producing CT images from attenuation coefficients measured by a CT detector assuming monoenergetic X-ray beam, ideal physics and geometry of the system, iterative reconstruction techniques use a multi-pass algorithm that additionally models real system geometry, X-ray beam statistics (different attenuation of parts of the polyenergetic X-ray spectrum), and encourages desirable image properties (smoothness, edges) (*Mehta et al., 2013*).

Unlike previous generations of iterative reconstruction techniques (statistical, hybrid) model-based solutions approach reconstruction as an iterative optimization process to find the "best fit" image to the acquired data, while penalizing the noise, through the use of data statistics, image statistics, and system models (*Mehta et al., 2013*). This results in greater reduction of the image noise, suppression of artifacts, improved spatial and low contrast resolution with greater scope for dose reduction while maintaining diagnostic image quality (*Hara et al., 2009*; *Mehta et al., 2013*; *Lambert et al., 2015b*).

Even though the technical performance of IMR has been validated early, the evaluation of diagnostic performance in specific applications unfolded gradually (*McCollough et al., 2009*; *Flicek et al., 2010*; *Lambert et al., 2015b*). In CT colonography (CTC), decreasing the radiation dose is even more important. Patients undergo CTC not only after incomplete optical colonoscopy (OC) or if colonic cancer is suspected, but also for primary screening (*Brenner & Georgsson, 2005*). Apart from reimbursement, radiation burden from CTC screening may be a concern because healthy individuals are exposed to radiation which is a weak carcinogen itself (*Albert, 2013*). So far, several papers on the technical performance of sub-milliSievert ultralow-dose (ULD) CTC have been published and there is limited information about its diagnostic performance and its improvement by IMR (*Lambert et al., 2015a*; *Lambert et al., 2015b*; *Nagata et al., 2015*; *Lubner et al., 2015*).

In this study, we compared the diagnostic performance of sub-milliSievert ULD CTC with standard dose (SD) CTC reconstructed with FBP, hybrid iterative reconstruction (HIR) and IMR techniques in the detection of colonic and extracolonic lesions.

## MATERIAL & METHODS

This prospective HIPAA compliant IRB approved study (reference number 1751/13 S/IV) was conducted in agreement with the Declaration of Helsinki and all patients signed an informed consent.

Between January 2014 and November 2014, 174 patients underwent CTC with two acquisitions per position where the standard dose was split in the proportion of 1:5. In 32 of them, at least one colonic lesion (colonic polyp $\geq 6$ mm in diameter or a colonic mass) was found. From the rest, another 32 age-, BMI-, and gender-matched patients with no colonic lesions were selected. The age of the patients was $67 \pm 12$ years and 42% were males.

CTC was performed after cathartic preparation with 200 mL of 40% magnesium sulfate in the evening, stool tagging with 250 mL 2.1% barium (Micropaque CT; GUERBET, Roissy, France) in the morning, noon and afternoon and dietary restriction on the day prior to the examination. Spasmolytic (butylscopolamine, Buscopan®, Boehringer Ingelheim, Germany) was administered to 94% of patients (4 patients had contraindications).
Insufflation of the colon by carbon dioxide was achieved by using a dedicated insufflator (PROTOCO$_2$L; Bracco Diagnostics Inc., Cranbury, NJ, USA).

The patients were scanned twice, both in the supine and prone positions at end-inspiration an on iCT Brilliance CT scanner (Philips Healthcare, Best, The Netherlands) with the following parameters in both positions: peak tube voltage 120 kV, planned tube time current product 50 mAs for SD acquisition and 10 mAs for ULD acquisition, detector collimation 128 × 0.625 mm, rotation time 0.5 s, pitch 0.601, and with current modulation (DoseRight[TM]). The images were reconstructed in 0.9 mm sections using a soft reconstruction kernel (filter A) for FBP and HIR (iDose[4]) set on the maximum level (level 6), and a routine body IMR level 2 (level 1 = weak, level 3 = strong). IMR is currently the latest commercially available generation of iterative reconstruction by the scanner manufacturer. All pairs of datasets were anonymized and transferred to a client workstation (Philips Intellispace Portal) with a dedicated CT colonography package and computer aided detection (CAD).

The images were reviewed by two independent readers with experience in reading CTC (>1,300 and >800 cases, respectively). The blinded studies were reviewed in a random order during a span of 6 months to minimize recall bias (*Pickhardt et al., 2012*). Colonic findings were primarily assessed in endoluminal or fillet view with CAD as a concurrent reader (*Choi et al., 2011*). A difference in identification of polyps between the two observers that occurred in five patients was resolved by consensus. The size of the polyps was measured in the endoluminal view and diminutive polyps (<6 mm in diameter) were not reported (*Pickhardt et al., 2008a*).

Both readers assessed image quality (IQ) of the colon in virtual endoscopic/fillet view and in 2D view (thin sections), and of the left adrenal gland (5 mm section thickness) on a five point Likert scale (1 = excellent, 5 = unevaluable), and reported colonic and extracolonic lesions according to daily practice and C-RADS classification (*Zalis et al., 2005*; *Lambert et al., 2015a*; *Lambert et al., 2015b*). The preferred endoluminal rendering threshold (HU value above which a voxel is rendered as colonic wall instead of intraluminal air) and the number of CAD marks were recorded as well. Image noise expressed as a standard deviation of Hounsfield density was measured by a technologist in identical parts of all six colonic segments (rectum, sigmoid, descending, transverse, ascending colon and cecum), in the aorta and in the left psoas muscle at the level of the fifth lumbar vertebra using a fixed region of interest.

The radiation dose was estimated from the dose length product multiplied by a weighting factor of 15 μSv/mGy cm (*Christner, Kofler & McCollough, 2010*) and in seven patients also using ImPACT CT Patient Dosimetry Calculator (ImPACT, London, UK).

Statistical evaluation was performed in Prism (Graphpad Software Inc., La Jolla, CA, USA) and R (The R Foundation for Statistical Computing, Vienna, Austria). We used the Friedman test with Dunns post hoc tests to compare the acquisitions and reconstruction algorithms. An exact binomial test was used to compare sensitivity and specificity. Interobserver agreement was expressed as Goodman and Kruskal's gamma statistics. SD acquisition served as the reference standard. A *P*-value below 0.05 was considered significant.

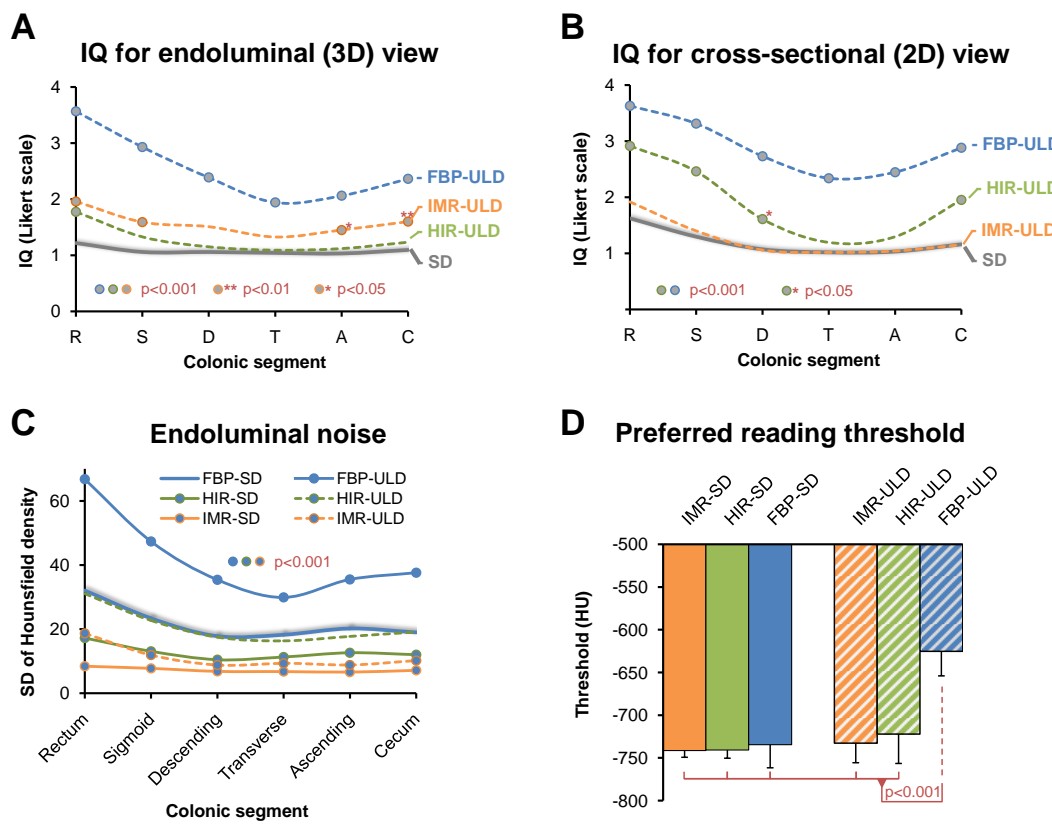

**Figure 1** Image quality (IQ) ratings for endoluminal (A) and cross-sectional (B) view for each colonic segment (1 = excellent, 5 = unevaluable) compared to the standard dose (SD) acquisitions (grey), which are represented as average from FBP-SD, HIR-SD, IMR-SD, show superiority of both iterative reconstruction algorithms compared to FPB in ultralow-dose CT colonography. Endoluminal noise measured as standard deviation of Hounsfield density in colonic lumen is suppressed with IMR-ULD, IMR-SD, and HIR-SD compared to FBP-SD (C). Statistical difference per segment is marked by circles. The preferred endoluminal rendering threshold, i.e., Hounsfield density that discriminates voxels representing intraluminal air from the colonic wall was significantly decreased in FBP-ULD indicating the need to suppress excessive noise (D). FBP, filtered back projection; HIR, hybrid iterative reconstruction; IMR, iterative model reconstruction technique; SD, standard dose; ULD, ultralow-dose.

## RESULTS

The average BMI of patients was $26.6 \pm 4.8$ kg/cm$^2$ and the mean radiation dose estimate was $4.1 \pm 1.4$ mSv for SD and $0.86 \pm 0.17$ mSv ($p < 0.0001$) for ULD for both positions.

The endoluminal noise per colonic segment, image quality in the virtual endoscopic and 2D view, preferred endoluminal rendering threshold, and clinical images are shown in Figs. 1–4. There were 27 small (6–9 mm) and 17 large ($\geq$10 mm) colonic lesions that were classified as sessile polyps ($n = 38$), flat lesions ($n = 3$), or as a mass ($n = 3$). The detection rate was lower for ULD-FBP compared to other reconstruction techniques ($p = 0.020$) and there were also more false positive results ($p = 0.011$, Fig. 5). Per patient sensitivity and specificity were 0.82 (95% CI [0.66–0.93], $p = 0.031$) and 0.93 (0.76–0.99, $p = 0.5$) for ULD-FBP, 0.97 (95% CI [0.83–1.0], $p = 1.0$) and 0.97 (0.80–1.0, $p = 1.0$) for ULD-HIR, 0.97 (95% CI [0.83–1.0], $p = 1.0$) and 1.0 (0.85–1.0, $p = 1.0$) for ULD-IMR. Per polyp
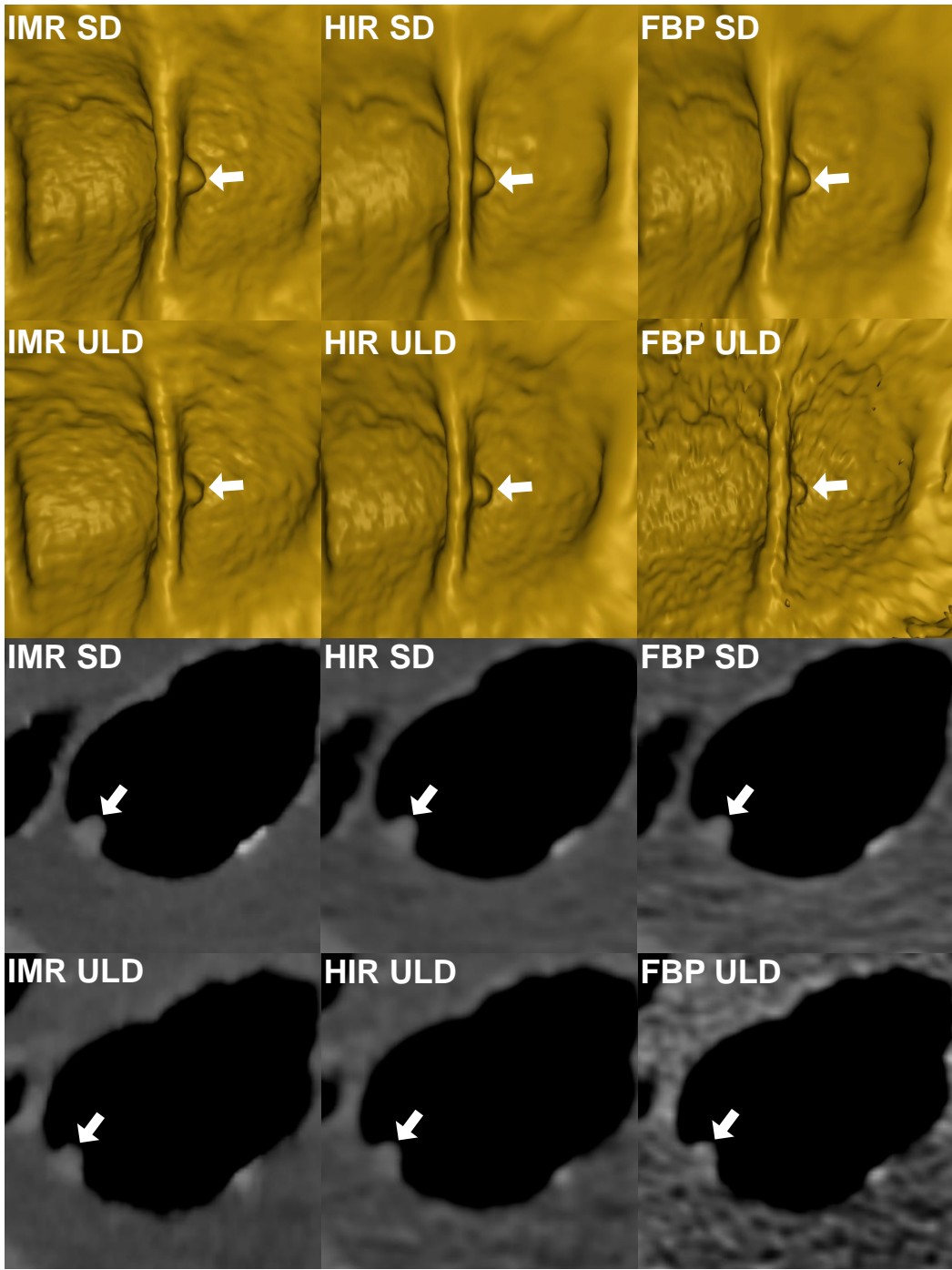

**Figure 2** Comparison of image quality of virtual endoscopic view and thin 0.9 mm sections in a colonic window (900/100 HU) of a small (8.2 mm) sessile polyp (arrow) in cecum shows markedly reduced image quality in ultralow-dose acquisition reconstructed with FBP. FBP, filtered back projection; HIR, hybrid iterative reconstruction; IMR, iterative model reconstruction technique; SD, standard dose; ULD, ultralow-dose.

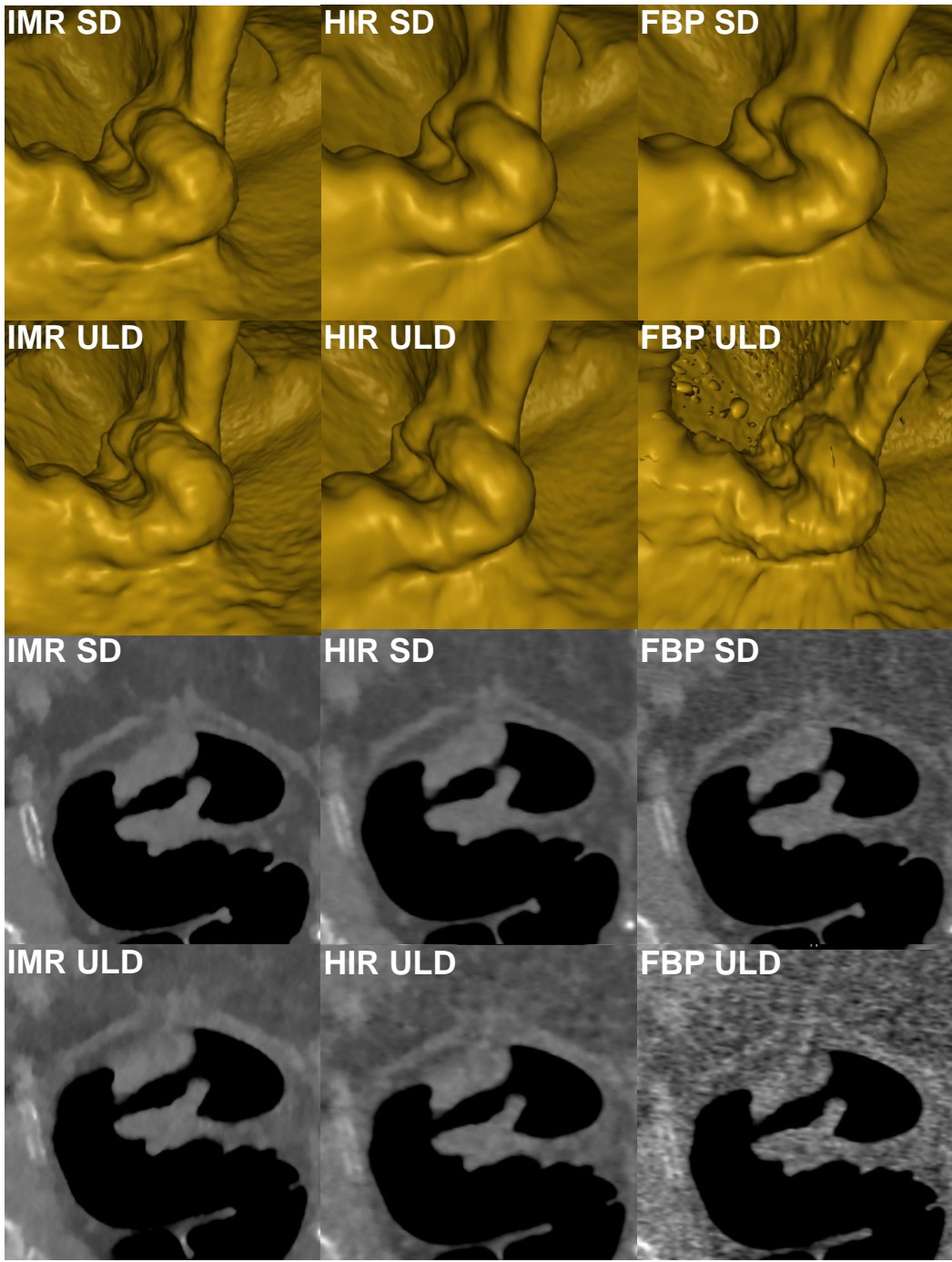

**Figure 3** **Comparison of image quality of a rectosigmoid tumor in virtual endoscopic view and thin 0.9 mm sections in a colonic window (900/100 HU) demonstrates markedly reduced image quality in FBP-ULD.** FBP, filtered back projection; HIR, hybrid iterative reconstruction; IMR, iterative model reconstruction technique; SD, standard dose; ULD, ultralow-dose.

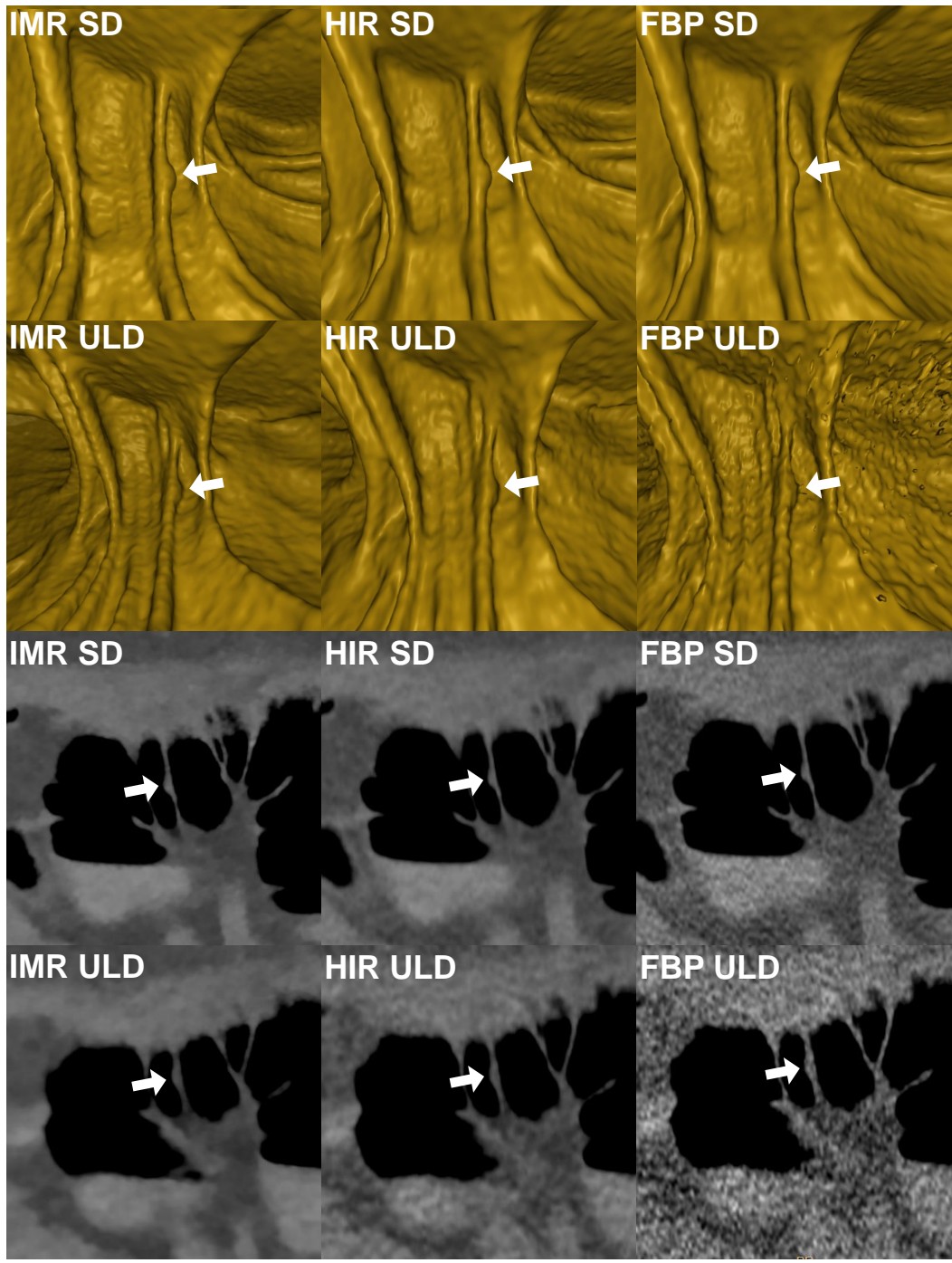

**Figure 4** **Comparison of image quality of a small (7.9 mm) flat lesion in the ascending colon shown in virtual endoscopic view and thin 0.9 mm sections in a colonic window (900/100 HU) demonstrates markedly reduced image quality in FBP-ULD.** FBP, filtered back projection; HIR, hybrid iterative reconstruction; IMR, iterative model reconstruction technique; SD, standard dose; ULD, ultralow-dose.

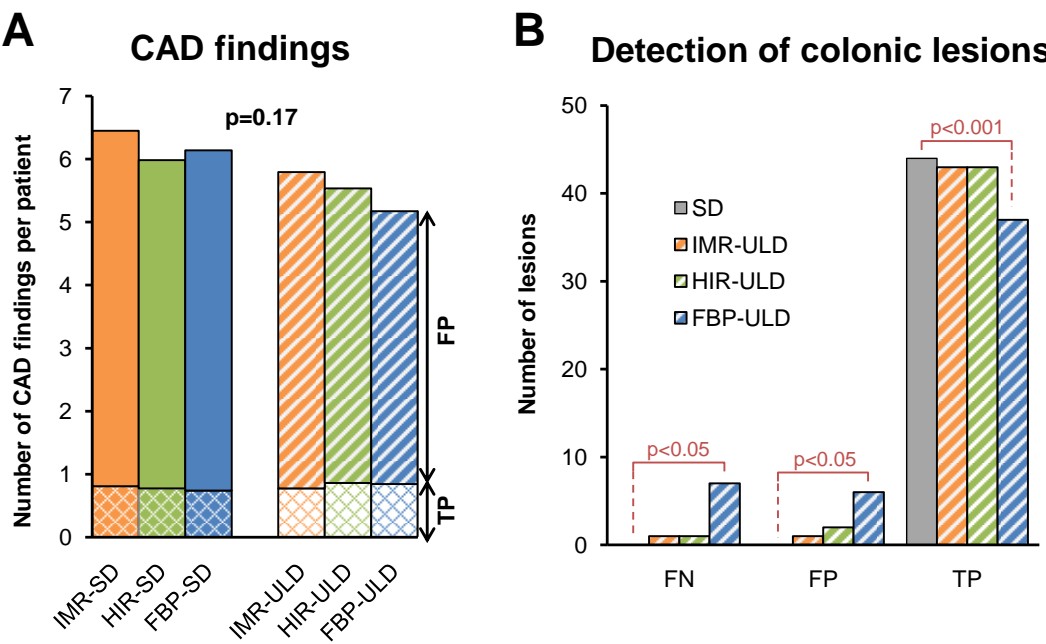

**Figure 5** Number of CAD findings represented as cumulative values for supine and prone acquisition (A). True positive (TP, cross-hatched pattern) and false positive CAD marks (FP, diagonal pattern) are distinguished. Detection of colonic lesions in ultralow-dose acquisitions (ULD) compared to standard dose acquisitions (SD) was reduced in FBP, which also had the greatest number of false positive (FP) and false negative (FN) findings (B). FBP, filtered back projection; HIR, hybrid iterative reconstruction; IMR, iterative model reconstruction technique; SD, standard dose; ULD, ultralow-dose; CAD, computer aided detection.

sensitivity was 0.84 (0.64–0.93, $p = 0.016$) for ULD-FBP, 0.98 (0.88–1.0, $p = 1.0$) for ULD-HIR, and 0.98 (0.88–1.0, $p = 1.0$) for ULD-IMR. In the local colonoscopy database, we found that 23 lesions in 16 patients were verified, the rest of the patients underwent colonoscopy elsewhere, or was scheduled for follow-up, or the findings were deemed unimportant by the physician.

There was no significant difference in the size and volume of polyps among all reconstruction techniques ($p = 0.077$ for size, $p = 0.49$ for volume). There were significantly less extracolonic findings detected in ULD-FBP and ULD-HIR, but in the E4 category (potentially important findings), the detection was similar (Fig. 6). The image noise in the aorta and in the left psoas muscle and the image quality of the left adrenal gland are shown in Fig. 6. The approximate reconstruction times were 40 s per position for FBP, 60 s for HIR, and 80 s for IMR.

The interobserver agreement for image quality of the virtual endoscopic, 2D view, and the left adrenal gland was 0.91, 0.90, and 0.83, respectively.

## DISCUSSION

Model based reconstruction is now commercially available in CT scanners of major vendors, who promise up to 80% reduction of the radiation dose while maintaining image noise and resolution (*Mehta et al., 2013*; *Löve et al., 2013*). CTC is one of

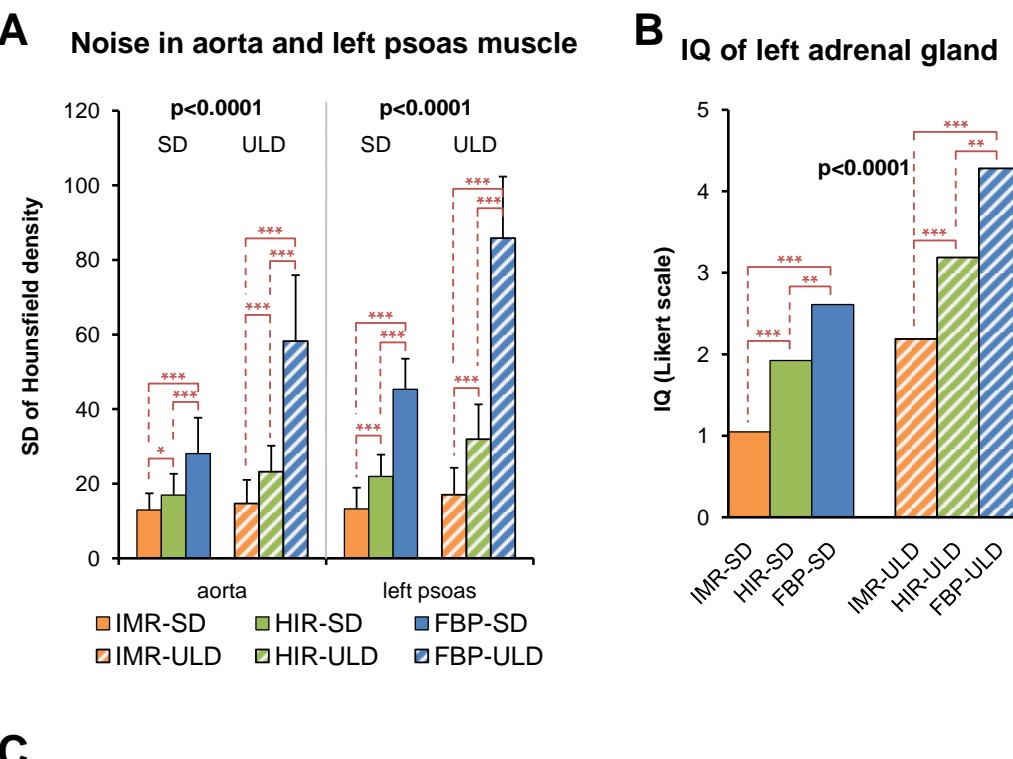

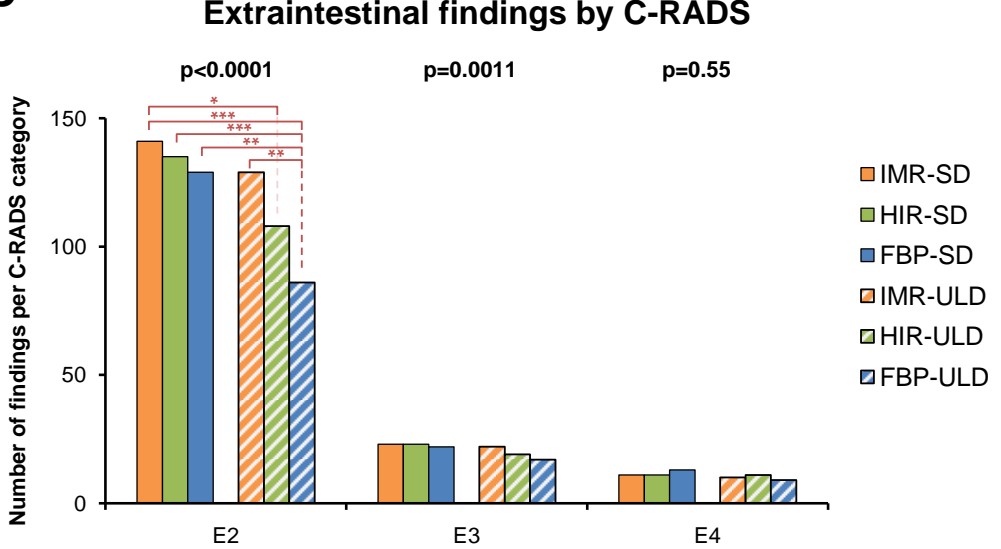

**Figure 6** Image quality of extracolonic structures represented by image noise in the aorta and the left psoas muscle (A) and rating of image quality (IQ) of the left adrenal gland (B, 1 = excellent, 5 = un-evaluable) demonstrate substantial decrease in the image noise especially in ULD with the IMR technique. Diagnostic performance for extracolonic findings grouped by C-RADS classification is reduced in the E2 category (unimportant findings) for low-dose acquisitions reconstructed with FBP and HIR (C). There is no difference in the E4 category (potentially important findings) and E3 category (likely unimportant findings, incompletely characterized). FBP, filtered back projection; HIR, hybrid iterative re-construction; IMR, iterative model reconstruction technique; SD, standard dose; ULD, ultralow-dose; *, $p < 0.05$; **, $p < 0.01$; ***, $p < 0.001$.

the applications where reducing the radiation dose is of major importance. It is increasingly available and used for screening of asymptomatic individuals (*Yee, 2013*; *Pickhardt, 2015*). The estimated benefit-to-risk ratio of CTC can be increased in direct proportion to dose reduction provided that polyp detection and discrimination among true polyps and polyp mimics remains unchanged. In case of sub-milliSievert ULD-CTC it can rise up to 209:1 (*De Gonzalez et al., 2011*).

In order to reduce radiation dose from CTC, several optimization strategies must be employed simultaneously (*Chang & Yee, 2013*). The radiation dose from CTC followed the same descending trend as other CT examinations. The majority of CTC studies in patients published in the noughties did not report the dose estimate, apart from a small number of papers that were addressing the radiation dose. First attempts to reduce the radiation dose in images reconstructed with FBP with a special noise-reducing filters resulted in dose estimates close to 2 mSv per study (*Iannaccone et al., 2003*; *Cohnen et al., 2004*). The iterative reconstruction technique that has been tested for CTC in the last five years, could decrease the dose even further, close to or even below 1 mSv for both positions altogether (*Lambert et al., 2015a*; *Lambert et al., 2015b*; *Lubner et al., 2015*). The introduction of size-specific dose estimate (SSDE), which was not used in this study is bound to even decrease the dose estimate in our slightly overweight patients but not on the absorbed dose *per se* (*Christner et al., 2012*; *Lubner et al., 2015*).

Although most studies in ULD-CTC reported unchanged polyp conspicuity that is closely related to their detection by both the human reader and CAD, it is also the ability to discriminate between polyps and polyp mimics (poorly tagged stool residuals, folds, inverted diverticula) especially in less distended segments, which makes an excellent reader and excellent CAD (*Fisichella et al., 2010*; *Lefere & Gryspeerdt, 2011*; *Pickhardt & Kim, 2013*). Unfortunately, the homogeneity of small polyps is difficult to measure because of the effect of partial volume averaging. Instead, surrogate parameters such as image noise in a different (larger) structure and subjective assessment of IQ are used to assess technical performance of the study. At the diagnostic level, this is reflected in false positive rather than false negative findings and in the fact that the performance of polyp detection declines more than the number of CAD marks, as also shown in this study (*Näppi & Yoshida, 2007*). In ULD-FBP, the IQ is reduced below an acceptable level which results in decreased sensitivity and specificity compared to HIR-ULD and IMR-ULD acquisitions where diagnostic performance approaches that of SD. The substantially decreased IQ of ULD-FBP is also reflected in its technical performance by increased image noise and poor ratings of IQ by the readers especially in the rectum and sigmoid colon. The perceived IQ in the endoluminal view can be to some extent improved by increasing the endoluminal rendering threshold, but this in turn results in a decreased size of the lesions that become less conspicuous (*Lambert et al., 2015b*).

The polyp size is an important biomarker of its position in the adenoma—carcinoma sequence (*Summers, 2010*). In this study, the polyp size and volume among SD and ULD did not vary significantly which means that HIR or IMR can be safely introduced without any correction of these measurements. There are other, more important variables influencing

polyp size such as distension, endoluminal rendering threshold and the viewing window (*Taylor et al., 2006*; *Summers, 2010*).

The detection of extracolonic pathology is considered one of the advantages of CTC over OC (*Pickhardt et al., 2008b*; *Badiani et al., 2013*). In this study, ULD examinations reconstructed with either FBP or HIR resulted in significantly lower detection of clinically unimportant findings (E2 category by C-RADS), which obviously had no clinical importance, and the number of potentially important findings (E4 by C-RADS) remained stable. In ULD-FBP, a lesion otherwise classified as "clinically unimportant" (E2) may turn into a "likely unimportant finding, incompletely characterized" due to limited visualization of its internal structure or increased density by excessive noise. This may result in the need of unnecessary workup and increased cost.

It has already been reported, that decreasing the radiation dose from CTC by half does not have any effect on polyp detection and lesion conspicuity, notably when iterative reconstruction is used, which makes the SD study acquired with 83% of the original dose (compared to the previous acquisition protocol) a valid standard of reference (*Flicek et al., 2010*; *Lubner et al., 2015*). The cumulative dose (about 5 mSv) was in line with what is currently done in the majority of institutions practicing CTC (*De Gonzalez et al., 2011*; *Albert, 2013*). The use of an additional ULD scan to assess its performance has already been reported in the literature (*Lubner et al., 2015*).

This study has several limitations. Firstly, the image appearance of different reconstruction techniques used in this study is well recognizable and therefore blinding of the studies may not have been effective enough. Secondly, we examined a solution by a single vendor, but other studies suggest that sub-milliSievert CTC is feasible with other CT scanners as well (*Flicek et al., 2010*; *Lubner et al., 2015*). Although the studies were reviewed in random order and with sufficient washout period, we cannot entirely exclude the effect of recall bias (*Pickhardt et al., 2012*). Since the predictive values are dependent on the prevalence of the disease according to Bayes' theorem, this study did not evaluate the negative and positive predictive values, due to the low prevalence of disease in the study population (18%). Because we use barium tagging that results in inhomogeneous opacification of intraluminal fluid, electronic cleansing is not used and therefore it was not tested. As the standard of reference, standard dose CTC was used, which has performance comparable to OC (*Pickhardt et al., 2003*).

In conclusion, this study showed that both hybrid and iterative model reconstruction techniques are suitable for sub-milliSievert ultralow-dose CT colonography without sacrificing the diagnostic performance of the study.

### Funding

This study was supported by the Ministry of Health No. RVO VFN 64 165 and the First Faculty of Medicine, Charles University in Prague (PRVOUK P27/LF1/1). The funders had no role in study design, data collection and analysis, decision to publish, or preparation of the manuscript.

## Grant Disclosures

The following grant information was disclosed by the authors:
Ministry of Health: RVO VFN 64 165.
First Faculty of Medicine, Charles University in Prague: PRVOUK P27/LF1/1.

## Competing Interests

Petr Ourednicek and Walter Giepmans are employees of Philips Healthcare. The other authors have no relevant financial relationships to disclose.

## Author Contributions

- Lukas Lambert conceived and designed the experiments, performed the experiments, analyzed the data, wrote the paper, prepared figures and/or tables, reviewed drafts of the paper.
- Petr Ourednicek performed the experiments, analyzed the data, contributed reagents/materials/analysis tools, wrote the paper, prepared figures and/or tables, reviewed drafts of the paper.
- Jan Briza analyzed the data, reviewed drafts of the paper.
- Walter Giepmans performed the experiments, contributed reagents/materials/analysis tools, reviewed drafts of the paper.
- Jiri Jahoda and Lukas Hruska performed the experiments, reviewed drafts of the paper.
- Jan Danes conceived and designed the experiments, reviewed drafts of the paper.

## Human Ethics

The following information was supplied relating to ethical approvals (i.e., approving body and any reference numbers):

Etická komise Všeobecné fakultní nemocnice v Praze (1751/13 S-IV).

"This prospective HIPAA compliant IRB approved study was conducted in agreement with the Declaration of Helsinki and all patients signed an informed consent."

## Data Availability

Raw numbers used for statistics have been uploaded as Supplemental Information.

## Supplemental Information

Supplemental information for this article can be found online at http://dx.doi.org/10.7717/peerj.1883#supplemental-information.

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
