# Peer review of "Sub-milliSievert ultralow-dose CT colonography with iterative model reconstruction technique"

_PeerJ, doi:10.7717/peerj.1883_

## Round 0.1 · original submission · Major Revisions

Dear Authors, as you can see the reviewers have pointed out several issues. Please respond to the questions and remarks for eventual re-review and evaluation.

Reviewer 1 ·

Basic reporting

This is a prospective HIPAA compliant IRB approved study in which the Authors compared the diagnostic performance of sub-miliSievert ULD CTC with standard dose CTC reconstructed with filtered back projection, hybrid iterative reconstruction and iterative model reconstruction techniques in the detection of colonic and extracolonic lesions.
The patients were scanned twice, both in the supine and prone positions with two acquisitions per position with a mean radiation dose of 4.1±1.4mSv for SD and 0.86±0.17mSv for ULD (for both positions).

Experimental design

The images were reviewed by two independent experienced readers.
The endoluminal noise per colonic segment, image quality in the virtual endoscopic and 2D view, preferred endoluminal rendering threshold, and clinical images.
Per patient sensitivity, specificity, positive and negative predictive values and per polyp sensitivity and positive predictive value were calculated.
They evidenced no significant difference in the size and volume of polyps among all reconstruction techniques.
The interobserver agreement for image quality of the virtual endoscopic, 2D view and the left adrenal gland was evaluated with good correlation.
They conclude that both hybrid and iterative model reconstruction techniques are suitable for sub-miliSievert ultralow-dose CT colonography without sacrificing the diagnostic performance of the study.

Validity of the findings

The study is well structured and analysis well performed. Statistical data are robust. Results encourages to think that ULD CTC sholud be performed in clinical practice.

Additional comments

-Please provide images of the lesions (sessile polyps, flat lesions, mass) of standard and low dose scans with all filters.
-Did you perform the prone and supine scan with the same parameters?
-Were there differences in results betweet ULD and SD scans in patient with incomplete preparation (colon cleansing, colon distension)?
-Please extend the text in analyzing the negative predictive value of the ULD scan with iterative reconstruction kernel.

Reviewer 2 ·

Basic reporting

MANUSCRIPT
INTRODUCTION
“So far, several papers on the technical performance of sub-miliSievert ultralow-dose (ULD) CTC have been published and there is limited information about its diagnostic performance and its improvement by IMR” (5,8). The authors cited only two references, however, other articles have been published recently; please, add the following articles in the references’ list:
1. Nagata K, Fujiwara M, Kanazawa H, et al. Evaluation of dose reduction and image quality in CT colonography: comparison of low-dose CT with iterative reconstruction and routine-dose CT with filtered back projection. Eur Radiol. 2015;25(1):221-9.
2. Lambert L, Ourednicek P, Jahoda J, et l. Model-based vs hybrid iterative reconstruction technique in ultralow-dose submillisievert CT colonography. Br J Radiol. 2015 Apr;88(1048):20140667.

- I suggest to include a bit of background regarding the novel iterative reconstruction techniques, explaining the differences between the hybrid iterative reconstruction (HIR) and the model-based iterative reconstruction (MBIR) techniques.

Experimental design

MATERIAL AND METHODS
“(>1300 and >800 cases).” please, change it into: (>1300 and >800 cases, respectively).
- Is the HIR iDose4 the latest generation iterative reconstruction technique of the Philips scanner? If yes, add this in the text.
- Please specify the six colonic segments used for assessment of the image noise.

Validity of the findings

RESULTS
- “The average BMI of patients was 26.6±4.8kg/cm2 and the mean radiation dose estimate
was 4.1±1.4mSv for SD and 0.86±0.17mSv for ULD for both positions.” please, report the p statistical significant value.
- I suggest to add a deeper explanation of Figures 1, 4 and 5 in the text and to highlight the results by reporting the p value for comparison.
- The detection rate was lower for ULD-FBP compared to other reconstruction techniques and there were also more false positive results (Fig. 4).” please, report the p statistical significant value.
- Please, add a test that compare the accuracy of the different reconstruction methods (ULD-FBP, ULD-HIR and ULD-IMR) and report the p statistical value.
- “but in the E4 category,” please, specify the meaning of the acronym “E4” for the extra-colonic findings (e.g. potentially important finding (E4 category)).
- About the interobserver agreement, “0.91, 0.90, and 0.83.” please, add: “0.91, 0.90, and 0.83, respectively.”

DISCUSSION
- “who promise upto” please, correct the typing error.
- “applications, where” please cut the comma.
- “It is obvious that the calculated benefit to risk ratio can be increased in direct proportion to dose reduction provided that polyp detection and discrimination among true polyps and polyp mimics remains stable, in case of sub-miliSievert ULD-CTC up to 209:1.” this statement needs to be rephrased and organized in a more simple and fluid way.
- “As the standard of reference, standard dose CTC was used, which has performance comparable to OC.” this statement needs a reference.

REFERENCES
References need a revision; there are some typing errors (in particular, check reference n.1, n.3, n.5).
Please, change the reference format according to the Journal style: “Each journal reference should be listed using this format: the full list of Authors with initials. Publication year. Full title of the article. Full title of the Journal, volume: page extents. DOI (where you have it). Example journal reference: Smith JL, Jones P, Wang X. 2004. Investigating ecological destruction in the Amazon. Journal of the Amazon Rainforest 112:368-374. DOI: 10.1234/amazon.15886.”

FIGURES
- Please add in the Figure Legend the definition of all the acronyms used in the figure.
-Figure 1. What do the asterisks mean in the bar-graphs of the “preferred reading threshold” (bottom right)?
- Figure 1, 4 and 5. For a better comprehension of the results obtained, I suggest to include a two-tailed p value of each iterative reconstruction method (ULD-FBP, ULD-HIR and ULD-IMR) in comparison with the standard-dose acquisitions; maybe it would be sufficient to just highlight the significant p value in the Figures.

Additional comments

There are some published reports that evaluated the benefit of ultralow-dose CT colonography using the iterative reconstruction methods compared to standard-dose imaging reconstructed with FBP (reference standard).
The article would strengthen this body of evidence, however, it needs some revision and integration of the statistical data; the authors assessed and compared the diagnostic performance of the different reconstruction methods but the comparison of data is incomplete.

---

## Round 0.2 · Minor Revisions

Some points of the 2nd reviewer have to be addressed,

Reviewer 1 ·

Basic reporting

The changes result in an overall improvement of the manuscript.

Experimental design

The images have been reviewed and all types of lesion are shown.
They conclude that both hybrid and iterative model reconstruction techniques are suitable for sub-miliSievert ultralow-dose CT colonography without sacrificing the diagnostic performance of the study.

Validity of the findings

Results encourages to think that ULD CTC sholud be performed in clinical practice.

Reviewer 2 ·

Basic reporting

No comments.

Experimental design

No comments.

Validity of the findings

No comments.

Additional comments

The manuscript was appropriately revised by the authors, that rigorously addressed each points raised by the reviewers. However, some points still need to be revised.
Please, find enclosed the PDF file with the annotations to the manuscript PDF.

Annotated reviews are not available for download in order to protect the identity of reviewers who chose to remain anonymous.

---

## Round 0.3 · accepted · Accept

The remaining reviewers' comments have been addressed and answered.

---

## Author Rebuttal · Round 0.3

**Dear Editor,**

We thank you and the undisclosed reviewers for reviewing our manuscript and for the favorable decision. We revised the manuscript according to the comments of the second reviewer:

**Reviewer 1 (Anonymous) - Comments for the author**

Thank you

**Reviewer 2 (Anonymous) - Comments for the author**

Briefly, we addressed all issues highlighted in the pdf in accordance with your recommendations. Thank you for improving the quality and consistency of our article very much.

ABSTRACT
page2#31: the p-value was added
page2#35: the technical performance data was added to the results section
page2#36: the inconsistency of this statement was corrected in Fig. 6.

BODY
INTRODUCTION
page3#52: Corrected

MATERIALS AND METHODS
page5#108: Corrected
page5#117: Corrected

RESULTS
page6#133: Corrected
page6#143: The inconsistency of this statement was corrected in Fig. 6.

DISCUSSION
page8#195: same as previous: The inconsistency of this statement was corrected in Fig. 6.
page9#217: Thank you. We took the liberty to use the wording that you suggested.

FIGURE LEGENDS
page15#322 (D): *** in D explained now in the chart
page15#324: Corrected
page15#328: IMR in general (solid and dashed orange lines in Fig. 1C) and even HIR-SD to be precise – it is now corrected.
Page21#359: asterisks now explained in the chart (b)
Page22#364: Corrected
Page22#365: Corrected

Page22#365: Corrected
Page22#367: Corrected both in text and in graph 6c.

Thank you again for choosing our manuscript for publication and improving its quality. Should there be any further suggestions, we would be delighted to address them.

On behalf of the authors, sincerely,
Lukas Lambert